# Effects of Building Directions on Microstructure, Impurity Elements and Mechanical Properties of NiTi Alloys Fabricated by Laser Powder Bed Fusion

**DOI:** 10.3390/mi14091711

**Published:** 2023-08-31

**Authors:** Shuo Wang, Xiao Yang, Jieming Chen, Hengpei Pan, Xiaolong Zhang, Congyi Zhang, Chunhui Li, Pan Liu, Xinyao Zhang, Lingqing Gao, Zhenzhong Wang

**Affiliations:** 1Luoyang Ship Material Research Institute, Luoyang 471023, China; 18220580820@163.com (S.W.); jiemingch@163.com (J.C.); panhengpei@163.com (H.P.); congyi610@163.com (C.Z.); 18437955231@163.com (C.L.); liupan_lp@163.com (P.L.); 13525971194@163.com (X.Z.); gaolq725@163.com (L.G.); jindou8309@163.com (Z.W.); 2Key Laboratory of Bionic Engineering, Ministry of Education, Jilin University, Changchun 130022, China; 3Henan Key Laboratory of Technology and Application Structural Materials for Ships and Marine Equipments, Luoyang 471023, China

**Keywords:** laser powder bed fusion (LPBF), NiTi SMAs, microstructure, mechanical response, building orientation, impurity elements

## Abstract

For NiTi alloys prepared by the Laser Powder Bed Fusion (LPBF), changes in the building directions will directly change the preferred orientation and thus directly affect the smart properties, such as superelasticity, as well as change the distribution state of defects and impurity elements to affect the phase transformation behaviour, which in turn affects the smart properties at different temperatures. In this study, the relationship between impurity elements, the building directions, and functional properties; the effects of building directions on the crystallographic anisotropy; phase composition; superelastic properties; microhardness; geometrically necessary dislocation (GND) density; and impurity element content of NiTi SMAs fabricated by LPBF were systematically studied. Three building directions measured from the substrate, namely, 0°, 45° and 90°, were selected, and three sets of cylindrical samples were fabricated with the same process parameters. Along the building direction, a strong <100>//vertical direction (VD) texture was formed for all the samples. Because of the difference in transformation temperature, when tested at 15 °C, the sample with the 45° orientation possessed the highest strain recovery of 3.2%. When tested at the austenite phase transformation finish temperature (Af)+10 °C, the 90° sample had the highest strain recovery of 5.83% and a strain recovery rate of 83.3%. The sample with the 90° orientation presented the highest microhardness, which was attributed to its high dislocation density. Meanwhile, different building directions had an effect on the contents of O, C, and N impurity elements, which affected the transformation temperature by changing the Ni/Ti ratio. This study innovatively studied the impurity element content and GND densities of compressive samples with three building directions, providing theoretical guidance for LPBFed NiTi SMA structural parts.

## 1. Introduction

NiTi shape memory alloys (SMAs) have received widespread attention from the materials science and engineering fields because of their excellent superelasticity, shape memory effect, damping properties, and biocompatibility. NiTi SMAs have a wide range of applications in aerospace, biomedical devices, mechanical electronics, the construction industry, and daily life [1,2]. Normally, NiTi SMAs prepared by traditional vacuum arc induction melting and powder metallurgy methods have homogeneous structures. However, thermal processing methods, such as drawing, forging, rolling, extrusion, and welding, are prone to introducing impurities, such as oxygen, carbon, and nitrogen, making the phase transformation temperature difficult to regulate. Frenzel et al. [3] pointed out that for the ratio of Ni and Ti elements, even a slight change of 0.1 at.%, can change their martensite transformation start temperature (Ms) by up to ~10 °C, thus significantly affecting their microstructure and mechanical properties, which greatly limits the application of NiTi SMAs in complex working conditions [4,5].

Laser powder bed fusion (LPBF) has a wide range of applications and has been widely used in a variety of materials, such as steel, high-temperature alloys, titanium alloys and aluminium alloys [6,7,8,9]. LPBF technology can transform complex three-dimensional shapes into two-dimensional layers and fabricate the desired shapes in one process. Its tunability in the process can solve the limitation of the phase transformation temperature, which is difficult to regulate during the traditional preparation and processing of NiTi SMAs [10,11]. Furthermore, LPBF also has great potential for the manufacturing of gradient materials. For example, Song et al. [12] proposes an analytical approach to design stretching-dominated truss lattices with tailored elastic properties, including isotropic elasticity, tailored zero/negative Poisson’s ratios, tailored Young’s moduli ratios along specified directions, and the prototype was prepared using the Micro-LPBF technique, these techniques could be used to develop new engineering applications and promote the development of NiTi SMAs [13,14,15]. Therefore, the NiTi SMAs fabricated by LPBF have become a major topic in the research community recently [16,17,18,19]. In these studies, the mechanical properties of LPBFed NiTi SMAs showed an obvious orientation dependence. For example, the type of defects has a great influence on the tensile strength, and the building directions have a significant effect on the wear properties of LPBFed NiTi SMAs [20,21]. Dadbakhsh et al. [22] investigated the anisotropy of LPBFed NiTi SMAs (with different building directions) and clarified that the crystallographic textures have an important effect on the mechanical properties of LPBFed NiTi SMAs. Gu et al. [15] optimized the process parameters of LPBFed NiTi, an excellent shape recovery rate of 88.23% was achieved under the optimal parameters, and a shape-recovery rate of 96.7% was achieved under electrical actuation for a structure with a pre-compressed strain of 20%. Shi et al. [19] investigated the effects of crystallographic anisotropy on the microstructure, phase transformation and the tribological properties of NiTi shape memory alloys fabricated by LPBF and revealed how different LPBF-induced microstructures affect mechanical properties and wear properties. Most of the studies above focused on the effects of the building directions on the crystallographic texture, phase composition, and thermomechanical response. However, the effect of the building directions on the impurity elements and the magnitude of the dislocation densities in LPBFed NiTi SMAs has not been reported.

In this study, the microstructures, mechanical properties, and impurity element contents of LPBFed NiTi SMAs with three different building directions (0°, 45°, and 90°) were investigated. In addition, the relationship between the preferred orientations and the dislocation densities was analyzed. Furthermore, the influence of the defects in the samples with different building directions on the introduction of impurity elements was also investigated in detail.

## 2. Materials and Methods

### 2.1. NiTi Samples Fabrication by LPBF

Ni_50.8_Ti_49.2_ powder was prepared by Minatech Ltd. (Shenzhen, China) using the electrode induction-melting gas atomization (EIGA) technique. The main composition (wt.%) of the NiTi powder was determined to be 55.80 wt.% Ni, 0.0576 wt.% O, 0.0066 wt.% C, 0.0067 wt.% N, and balance Ti. Figure 1a shows the scanning electron microscopy (SEM) image of the NiTi powder, which exhibits a regular spherical shape with less satellite powder, and the size range of the powder particles is from 15 μm to 53 μm (D50 = 36.8 μm). As shown in Figure 1b, three sets of cylindrical specimens were prepared with a height of 10 mm and a diameter of 6 mm. Figure 1c shows the scanning strategy of 67° rotation angles between the adjacent layers. The LPBF processing was performed in a BLT (BLT S210, Shaanxi, China) machine equipped with a 500 W ytterbium-doped laser under argon protection to keep the oxygen level below 100 ppm. Previous research has shown that NiTi alloys exhibits favorable superelasticity and low porosity when exposed to the energy density of about 72 J/mm^3^. To control independent variables, the same process parameters and different building directions are applied. Table 1 shows the optimized parameters in the LPBF processing, and the energy density was calculated with E = P/vht [21].

### 2.2. Microstructure and Property Characterization

ADSC250 differential scanning calorimeter (DSC, TA) was used to determine the phase transformation temperature of the samples. The weight of the DSC specimens was 5~20 mg, and the heating/cooling rate was 15 °C/min from −80 °C to 70 °C. The LPBFed NiTi SMAs were ground, polished, and etched using a mixture of 70 vol% H_2_O + 20 vol% HNO_3_ + 10 vol% HF solution. The metallography of the samples was observed by an optical microscope (OM, Zeiss, Oberkochen, Germany). The atomic ratios of Ni and Ti were analyzed by scanning electron microscopy (SEM, FEI Scios 2, Waltham, MA, USA) equipped with an EDAX X-ray energy spectrometer (EDS). Five random points were selected to test each sample to obtain the average composition. The tested samples were polished to 2.5 μm with 180~2000 grit silicon carbide sandpaper, followed by electrolytic polishing in a HNO_3_/CH_3_OH = 1:10 (vol%) solution at 20 V for 15 s. The crystallographic orientation, phase composition and dislocation density were analyzed by SEM equipped with an EDAX electron backscatter diffraction (EBSD) system.

Transmission electron microscopy (TEM) samples were ground to a thickness of 50 μm and then electropolished using a twin-jet thinning electropolishing device and an electrolyte consisting of 4% perchloric acid and 96% ethanol (vol%) at −20 °C. TEM observation and electron diffraction analysis were performed in a JEM2100 (JEOL, Tokyo, Japan) electron microscope at 200 kV.

Sample impurity elements were collected by an ONH836 gas analyser and a CS800 carbon and sulphur analyser. Compression tests were performed in an INSTRON 8862 mechanical testing machine with a strain rate of 5 × 10^−4^/s. The microhardness was tested on a Wilson VH3300 (Buehler, Lake Bluff, IL, USA) microhardness tester. Among them, five positions were collected for each sample to obtain the average microhardness value.

## 3. Results and Discussion

### 3.1. Defects and Impurities

As shown in Figure 2, the OM images of the 0°, 45° and 90° samples demonstrate that the porosity defects represented by the black dots are distributed inside the grains and at the grain boundaries. It is noteworthy that the unmelted defects are concentrated at the melt pool boundaries. Specifically, the 0° samples have more unmelted defects but smaller sizes and fewer porosity defects. The 90° samples have more serious porosity defects and fewer unmelted defects, while the number and size of unmelted defects and porosity defects of the 45° sample are moderate. Actually, the difference in the building direction changes the defect distribution of the samples. During LPBF processing, different thermal histories result in different grain solidification directions and differences in convection, leading to differences in the distribution of unmelted defects and differences in microporous defects [21]. In addition, the spattering of the molten metal varies depending on the direction of solidification, leading to differences in the distribution of the defects [23,24,25,26].

Slight changes in the Ni/Ti ratios can cause drastic changes in the phase transformation temperatures [27,28]. Among them, the introduction of impurity elements could change the phase composition of the samples by forming the second phases, i.e., Ti_4_Ni_2_O_X_, TiN, TiO_2_ and TiC, which will change the transformation temperatures. The functional properties of the samples will change further [29,30]. Chemical element analysis was performed on the virgin powder and the samples. The results are shown in Figure 3, and the LPBF samples have a slightly higher oxygen content than the virgin powder. Among them, the 0° sample has the highest oxygen content, while the 45° sample has the lowest. The results are consistent with the analysis of the OM images, where the 0° sample has the highest number of defects, while the 45° sample has the lowest number of defects. In addition, the carbon and nitrogen contents of the as-fabricated samples are lower than those of the virgin powder, which indicates that the high temperature of the laser cladding process acts on the powder, resulting in the loss of carbon or nitrogen.

### 3.2. Microstructure Analysis

LPBF has the characteristics of directional temperature gradients and interlayer remelting. The grains could grow along the highest thermal gradient, and the competition between the grains for growth will also lead to the preferential growth of favourable crystallographic planes and favourable crystallographic directions during the processing of LPBF. Obviously, the building directions will significantly change the preferred orientation of the samples. Therefore, the texture of the LPBFed NiTi SMAs will be changed by the difference in the building directions in this work [31,32,33,34].

To clarify the texture of the samples with different building directions, EBSD analysis was performed. As shown in Figure 4a, the inverse pole figure (IPF) of the 0° sample along the RD-TD direction shows a clear solidification texture of <100>//TD along the VD. According to the pole figure of the 45° sample (Figure 4b), the {001} pole appears at 45° from the centre of the projection plane, indicating that the {001} texture is rotated by 45° compared to the 0° sample. Essentially, the {001} of the 45° sample is still growing parallel to the VD. As shown in Figure 4c, the 90° sample has a strong solidification texture of <100>//RD//VD. Interestingly, the pole figure of the 90° sample overlaps with that of the 0° sample by rotating the pole figure of the 90° sample by 90°, which indicates that the 90° sample has the same preferred orientation as the 0° sample. According to the analysis of the EBSD results, the preferred orientation of the LPBFed NiTi SMAs can be obviously changed by adjusting the building direction; meanwhile, a <100>//VD was formed for all samples with different building directions.

Figure 5 shows the phase composition, kernel average misorientation (KAM) and grain boundary distribution of samples with different building directions. As shown in Figure 5a,d,g, the martensite phase (shown in green) content of the 0°, 45° and 90° samples are 5.5%, 0.7% and 3.1%, respectively. Obviously, the martensite phase content of the 45° sample is lower than that of the others, which indicates that the martensite phase transformation temperature of the 45° sample is lower than that of the others. The KAM value is the statistic for calculating the overall average misorientation of the pixel points collected, as shown in Figure 5b,e,h. The KAM values of the three samples are basically the same, i.e., 0.73, 0.72 and 0.73, respectively. Moreover, the grain boundary distribution of different samples is shown in Figure 5c,f,i. Generally, the lattice distortion is stronger in the area of the martensitic phase. the area perpendicular to the melt pool that is parallel to the building direction. Meanwhile, as shown in the KAM maps, there is also a large lattice distortion at the high-angle grain boundary, which indicates that the grains at this location are small, and more nucleation sites accompanied by large heat flow gradients during the LPBF process have led to the formation of more fine grains.

Figure 6 shows the geometrically necessary dislocation (GND) densities of the 0°, 45° and 90° samples. The specific values calculated by EBSD are shown in Figure 6d. Among them, the GND densities of the 0°, 45° and 90° samples are 4.45 × 10^13^/m^2^, 3.89 × 10^13^/m^2^ and 3.84 × 10^13^/m^2^, respectively. In contrast, the 45° sample has the highest GND density, while the densities of the 0° and 90° samples were relatively low. Since the martensitic phase contains more dislocations and twin substructures, which lead to an increase in the overall dislocation density value, the limit of the misorientation value for calculating the GND density is set to 1.5°, excluding the influence of the martensitic phase.

To further analyze the effect of the building directions on the samples, TEM was used to observe the phase composition and phase distribution of different samples. As shown in Figure 7, a large number of Ti_2_Ni phases or Ti_4_Ni_2_O_X_ phases with sizes ranging from 50 to 100 nm can be observed. In addition, the lath-shaped B19′ martensite phase with sizes ranging from 100 to 300 nm can also be observed. Selected area electron diffraction was applied to identify the Ti_2_Ni/Ti_4_Ni_2_O_X_ phase and the B19′ martensite phase. Similarly, Figure 8 shows the distribution of dislocations, Ti_2_Ni phase and B19′ martensite phase of all the samples. Among them, the size of the Ti_2_Ni/Ti_4_Ni_2_O_X_ phase in the 0° and 90° samples is larger than that in the 45° sample. In addition, according to the TEM bright-field morphologies, the dislocation density of the 45° sample is higher than that of the 0° and 90° samples, while more martensite phases were observed in the 0° and 90° samples. Based on the phenomenon reflected by the TEM morphology above, we found that the statistical results of the GND density and martensite phase content are consistent with the EBSD analysis. TEM analysis proved that larger sized Ti_2_Ni/Ti_4_Ni_2_O_X_ phases were associated with higher oxygen content in the 0° and 90° samples and that samples with different building directions contained both Ti_2_Ni/Ti_4_Ni_2_O_X_ phases and martensite phase.

### 3.3. Phase Transformation Analysis

The reversible phase transformation between the austenite phase and martensite phase is the theoretical basis for the shape memory effect and superelasticity of NiTi alloys. Among them, the martensite start temperature (Ms), martensite finish temperature (Mf), austenite start temperature (As) and austenite finish temperature (Af) are the important phase transformation temperatures during the phase transformation process. Figure 9a shows the result of the DSC measurement of the three samples. The corresponding phase transformation temperatures obtained by the tangent method are listed in Table 2. The Ms values of the 0°, 45° and 90° samples are 22.1 °C, 11.4 °C and 13 °C, respectively. Obviously, the Ms of the 0° sample was the highest, while that of the 45° sample was the lowest. Correspondingly, the order of the martensite phase content in all samples at ambient temperature (15 °C) was 0° sample > 90° sample > 45° sample, which is consistent with the result of the EBSD analysis. For each sample, five EDS results were collected to calculate the average value, and the Ni content results are shown in Figure 9b. It can be seen that the higher the Ni content of the sample, the lower the phase transformation temperature, which is consistent with the DSC results.

The evaporation of Ni, the introduction of impurity elements, and the difference in thermal conductivity are the reasons for the difference in the phase transformation temperature. During the processing of LPBF, a higher energy input could lead to the evaporation of Ni, which in turn changes the Ni/Ti ratio in the NiTi alloys [30,35]. Nevertheless, the LPBF processing parameters were the same except for the building direction in this work, so the evaporation of Ni in different samples did not have a significant effect on the phase transformation temperature. Second, impurities are introduced, i.e., C, O, N, etc. Walker et al. [36] pointed out that the introduction of impurities will react with the matrix elements to form compounds, i.e., TiC, Ti_4_Ni_2_O_X_, TiO_2_, etc., which will affect the Ni/Ti ratios and further the phase transformation temperature. As shown in Figure 3, the results of the chemical test demonstrate that the content of oxygen is 0° sample > 90° sample > 45° sample. Obviously, the loss of Ni was greatest in the 0° sample, which led to a decrease in the Ni/Ti ratio and an increase in the phase transformation temperature. Last, the difference in thermal conductivity. As described above, the raw powder and LPBF parameters used in this work are the same. Therefore, it is considered that thermal conductivity has no significant effect on the phase transformation temperature. In conclusion, the difference in the content of impurity elements caused by different building directions is the most important factor causing the difference in phase transformation temperatures.

### 3.4. Mechanical Properties Analysis

Figure 10 shows the microhardness test results for the three samples, with the 45° sample having the highest microhardness and the 0° sample having the lowest microhardness. Generally, dislocation densities are an important factor for the microhardness of metallic materials. The increase in the dislocation density could lead to an increase in the matrix strength and hardness of the material. As described in Section 3.2, the 45° sample has the highest dislocation density, while the dislocation densities of the 0° and 90° samples were relatively low; therefore, as a result, the highest microhardness values were obtained for the 45° samples. In addition to the dislocation density, phase transformation is also an important factor affecting the microhardness. Stress-induced martensite phase transformation and martensite reorientation may occur during microhardness testing, which can lead to a change in the microhardness values [37]. Based on the phase transformation temperature (Table 2), we can see that the 0° sample has the highest phase transformation temperatures, followed by the 90° sample, while the 45° sample has the lowest. However, as shown in Figure 10, the trend of the microhardness values of the 0° sample, 45° sample and 90° sample are opposite to the phase transformation temperatures: 45° sample > 90° sample > 0° sample. A high phase transformation temperature is prone to induce the stress-induced martensite phase transformation, and martensite reorientation may occur with less difficulty. Therefore, the microhardness value of the 0° sample was the lowest.

Figure 11 shows the compression stress–strain curves. The recoverable strains and recovery ratios are shown in Table 3. As shown in Figure 11a, all the samples were subjected to a strain of 5% at 15 °C. Table 3 shows that the 45° sample has the highest recoverable strain of 3.2%, while the lowest recoverable strain is 1.8% for the 0° sample under the conditions above. As described in the phase transformation temperature section, the Ms of the 45° sample is the lowest compared with the others. Therefore, during the superelasticity test, the 45° sample with the highest proportion of the austenite phase shows the greatest driving force of recovery and the highest recoverable strain.

As shown in Figure 11b, the superelasticity was tested at the temperature of Af + 10 °C. The recoverable strains of the 0° and 90° samples were improved to 5.42% and 5.83%, respectively. Interestingly, the 90° sample has the best recoverable strain, while the recoverable strain of the 45° sample is the lowest. As discussed above, samples with different building directions have different textures along the loading direction. According to the result of the EBSD analysis, the loading direction of the 90° sample coincides with the <100> orientation. Generally, <100> is a hard orientation, which is not conducive to plastic deformation. Moreover, the 90° sample is more likely to drive stress-induced martensitic phase transformation behaviour without causing dislocation accumulation [38]. Therefore, the 90° sample has the best superelasticity compared to the others at the temperature of Af + 10 °C. The 45° sample has a <110> texture along the loading direction with a relatively large Schmidt factor, which is prone to plastic deformation. Therefore, the plastic deformation will remain in the matrix after unloading and cannot be recovered. Meanwhile, the low Ms of the 45° sample requires a larger stress to drive the martensitic phase transformation, resulting in plastic deformation without a significant stress-induced martensitic phase transformation platform.

Therefore, the superelasticity of the 45° sample is worst compared with the others at the temperature of Af + 10 °C.

A summary of the research work on LPBFed NiTi SMAs with superior superelastic properties (tested at Af + 10 °C) was performed, and the results are shown in Table 4. It can be seen that at smaller deformations, the samples generally have higher strain recovery rates, while larger strain is prone to have higher strain recovery, but the recovery rates are not ideal. The superelastic properties in this work are similar to those in similar studies.

## 4. Conclusions

In this study, LPBF-NiTi samples with different building directions (0°, 45° and 90°) were tested and analyzed for microstructure, defect distribution, impurity element, crystallographic orientation, dislocation density, phase transformation behaviour and mechanical properties using the same LPBF processing parameters, and the following conclusions were obtained:(1)The samples with different building directions exhibited distinct defect distributions. The 0° and 90° samples, which had more unmelted defects, demonstrated a higher oxygen content. The introduction of oxygen impurities altered the Ni/Ti ratios in the matrix and significantly raised the phase transformation temperature. The order of martensitic phase transformation temperature was as follows: 0° sample > 90° sample > 45° sample. This order is in line with the oxygen content present in each sample. Consequently, the quantity of martensite phase in samples with different building directions also followed the same trend: 0° sample > 90° sample > 45° sample.(2)The increase in dislocation density enhanced the microhardness of the matrix, while the increase in phase transformation temperature facilitated the occurrence of stress-induced martensitic phase transformation. This phenomenon was also the primary factor responsible for altering the hardness of LPBFed NiTi SMAs.(3)LPBFed NiTi SMAs with different building directions had different preferred orientations along the loading direction, but all had textures of VD//<100>.

The stress-induced martensitic phase transformation was related to crystallographic orientation. Unfavourable orientations can result in plastic deformation and obscure the platform of the stress-induced martensitic phase transition. When the phase transformation temperature (Af) exceeded the ambient temperature, it significantly impacted the superelastic properties, with higher temperatures leading to poorer superelastic properties.

The main idea of this study was to change the introduction of impurity elements in NiTi alloys and alter their microstructures by adjusting the LPBF building direction, which in turn affects their phase transformation temperatures and superelasticity. In conclusion, for engineering applications, the 45° sample had a lower phase transformation temperature, which was more favourable for superelastic recovery under certain temperatures (As < testing temperature < Af), while the preferred orientation of the 90° sample ensured a greater superelastic recovery strain when tested at Af + 10 °C.

## Figures and Tables

**Figure 1 micromachines-14-01711-f001:**
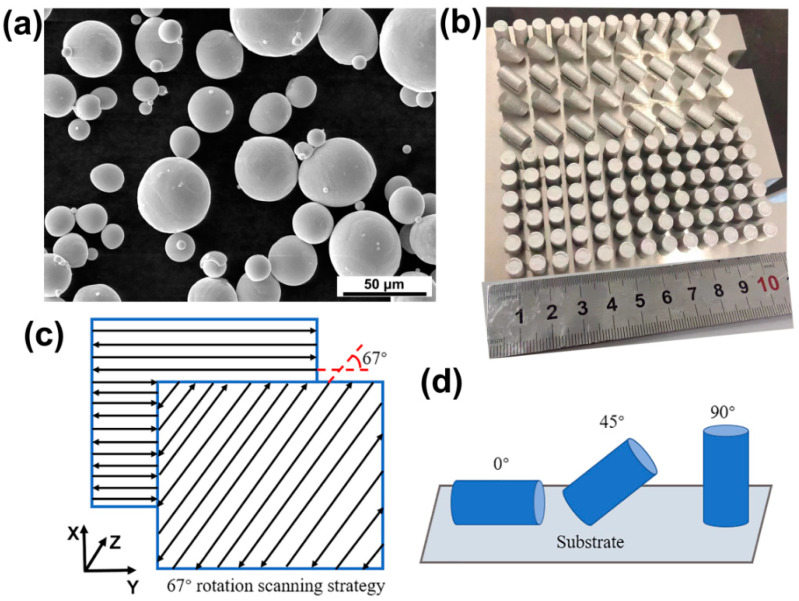
(**a**) Morphology of Ni_50.8_Ti_49.2_ powder, (**b**) actual view of LPBFed NiTi SMA samples, (**c**) scanning strategy implemented in this research and (**d**) schematic of LPBFed NiTi SMA samples.

**Figure 2 micromachines-14-01711-f002:**
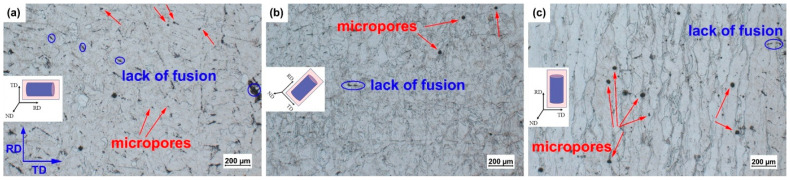
Optical microstructure of (**a**) 0°, (**b**) 45° and (**c**) 90° samples (the rolling direction (RD) and transverse direction (TD) planes are the observation planes).

**Figure 3 micromachines-14-01711-f003:**
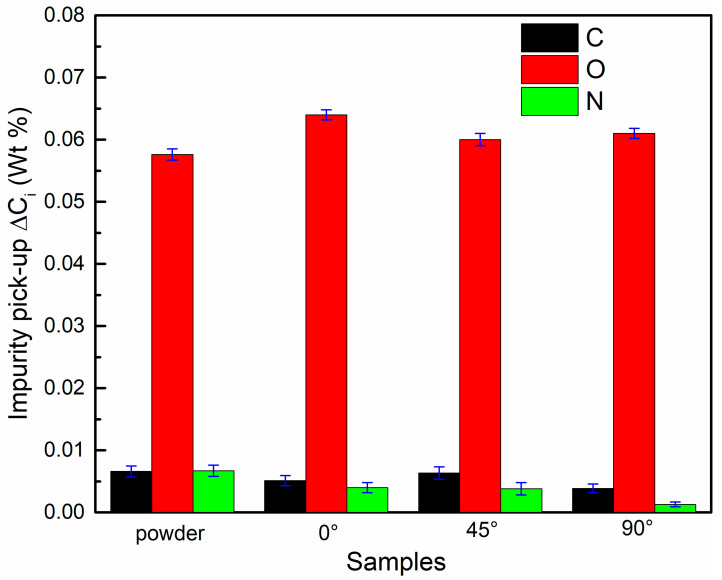
The contents of carbon, oxygen and nitrogen in the virgin powder and 0°, 45° and 90° samples.

**Figure 4 micromachines-14-01711-f004:**
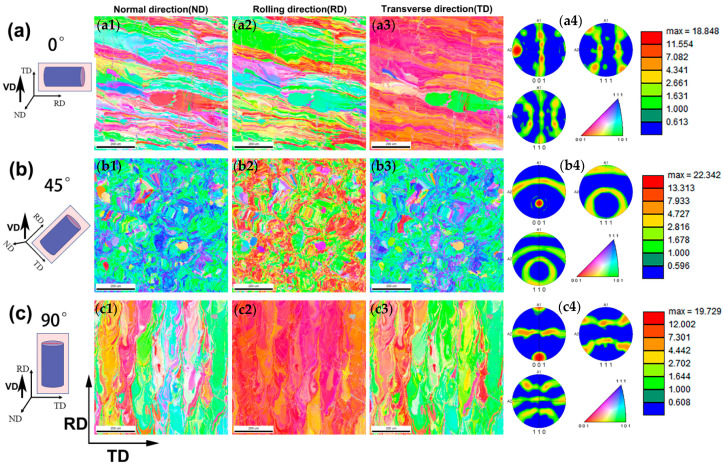
Schematic of the EBSD collection position for the (**a**) 0°, (**b**) 45° and (**c**) 90° samples: (**a1**–**c1**) are IPF orientation maps of LPBFed NiTi SMAs along the normal direction; (**a2**–**c2**) are IPF orientation maps of LPBFed NiTi SMAs along the rolling direction; (**a3**–**c3**) are IPF orientation maps of LPBFed NiTi SMAs along the transverse direction; and (**a4**–**c4**) are corresponding {001}, {110} and {111} pole figures of the 0°, 45° and 90° samples.

**Figure 5 micromachines-14-01711-f005:**
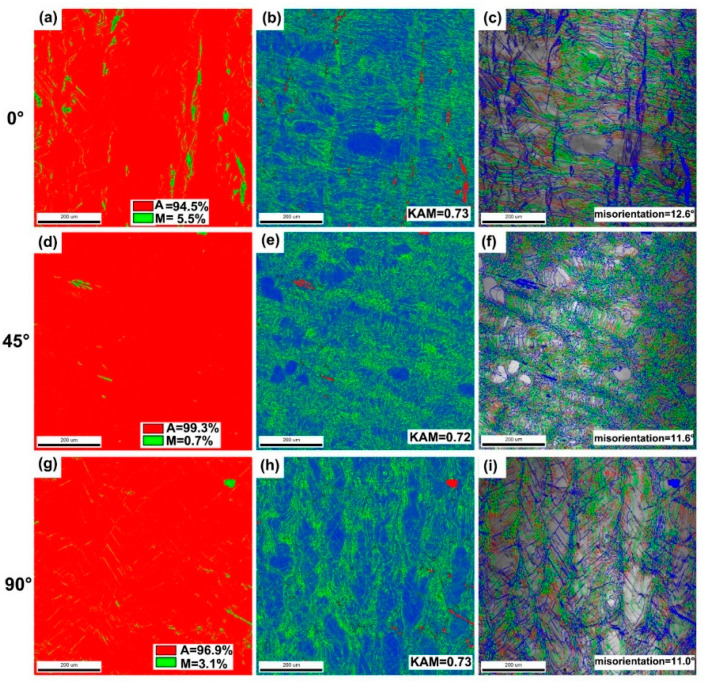
(**a**–**c**) EBSD analyses of the 0° sample. (**d**–**f**) EBSD analyses of the 45°sample. (**g**–**i**) EBSD analyses of the 90°sample. (**a**,**d**,**g**) Phase composition analyses. (**b**,**e**,**h**) KAM analyses. (**c**,**f**,**i**) Grain boundary distribution analyses.

**Figure 6 micromachines-14-01711-f006:**
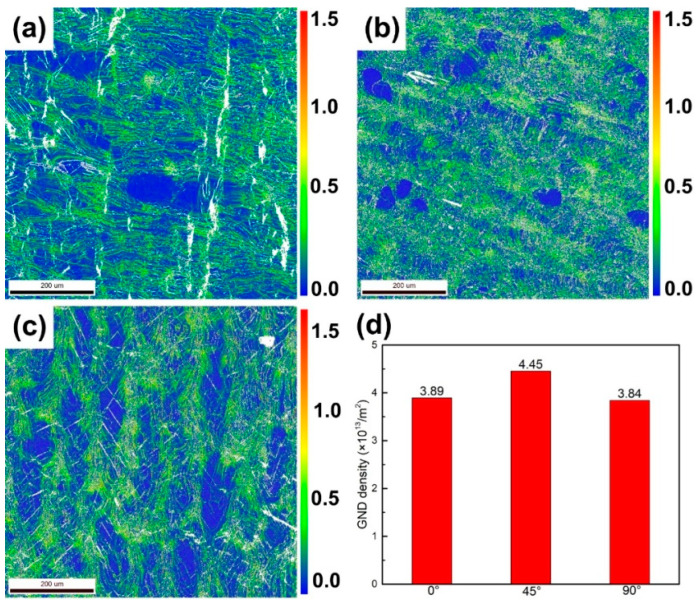
(**a**) GND distribution of the 0° sample, (**b**) 45° sample and (**c**) 90° sample and their corresponding (**d**) GND density values.

**Figure 7 micromachines-14-01711-f007:**
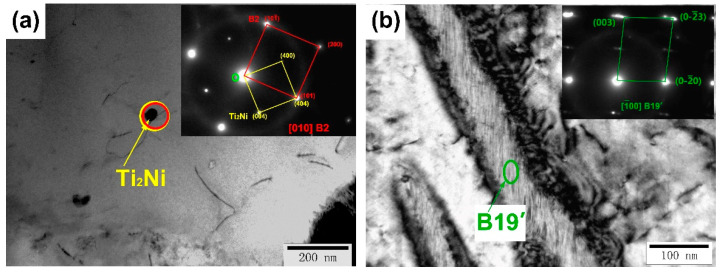
(**a**) Bright field image and selected area electron diffraction in the corresponding circle region for the 0° sample (matrix and Ti_2_Ni phase). (**b**) Bright field image and selected electron diffraction in the corresponding elliptic region for the 0° sample.

**Figure 8 micromachines-14-01711-f008:**
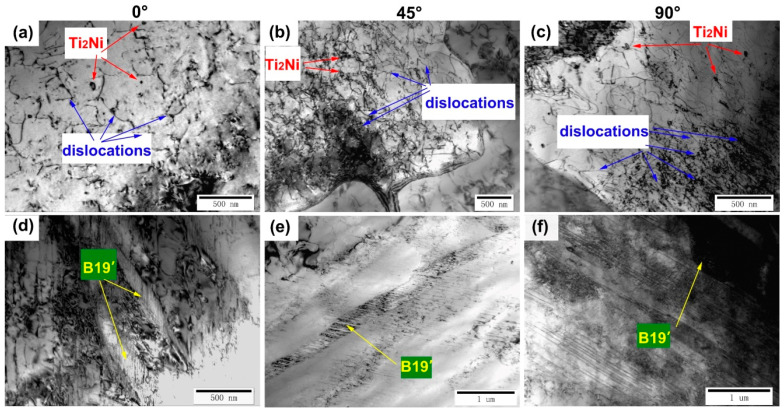
(**a**,**d**) TEM morphology of 0° samples. (**b**,**e**) TEM morphology of 45° samples. (**c**,**f**) TEM morphology of 90° samples.

**Figure 9 micromachines-14-01711-f009:**
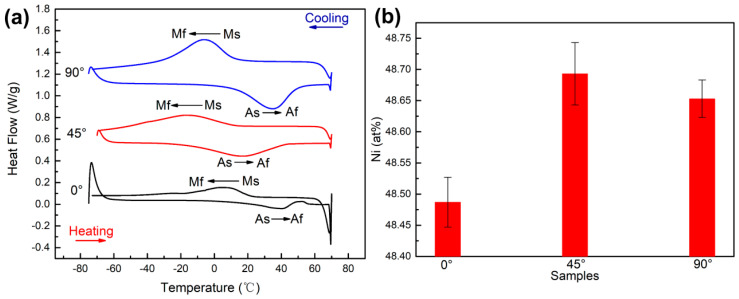
(**a**) DSC curves of 0°, 45° and 90° samples. (**b**) Corresponding Ni content of 0°, 45° and 90° samples.

**Figure 10 micromachines-14-01711-f010:**
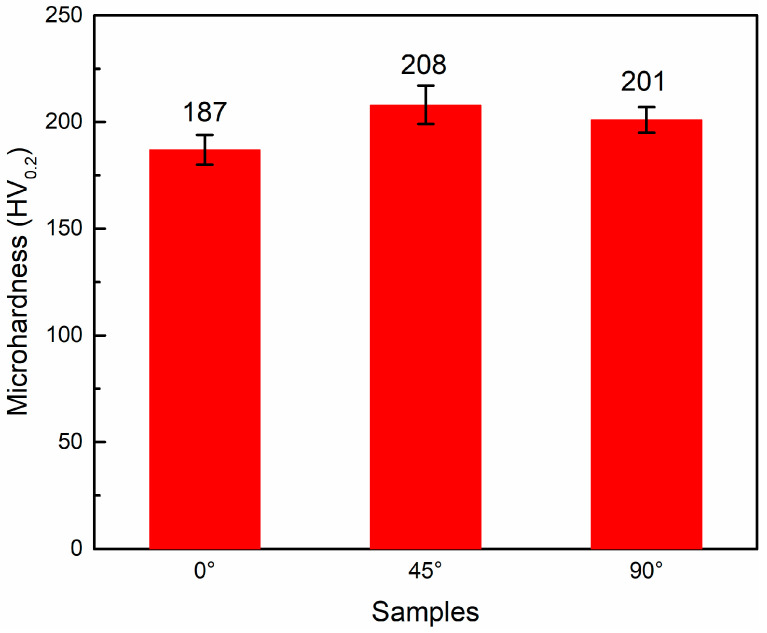
Microhardness of 0°, 45° and 90° samples.

**Figure 11 micromachines-14-01711-f011:**
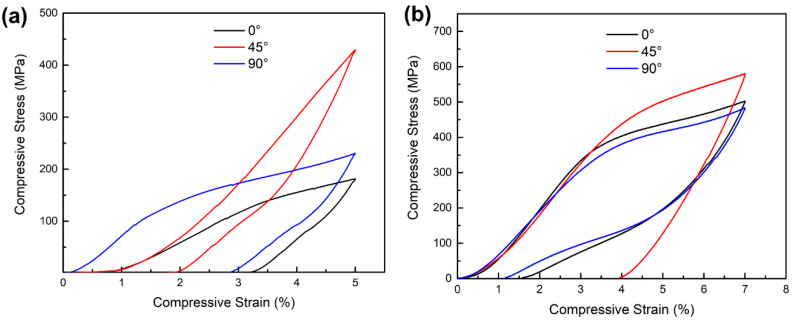
Compressive engineering stress-engineering strain curves for 0°, 45° and 90° samples (**a**) tested at ambient temperature (≈15 °C) and (**b**) tested at Af + 10 °C.

**Table 1 micromachines-14-01711-t001:** Processing parameters for LPBFed NiTi SMAs.

Sample	Laser Power P (W)	Scanning Speed v (mm/s)	Hatch Spacing h (μm)	Layer Thickness t (μm)	Energy Density E (J/mm^3^)
0°	105	600	80	30	72.92
45°	105	600	80	30	72.92
90°	105	600	80	30	72.92

**Table 2 micromachines-14-01711-t002:** Phase transformation temperatures of the 0°, 45° and 90° samples.

Sample	0°	45°	90°
Mf (°C)	−17.5	−54.2	−30.7
Ms (°C)	22.1	11.4	13
As (°C)	13.9	−15.9	11.1
Af (°C)	52.3	43.7	51

**Table 3 micromachines-14-01711-t003:** Summary of compressive superelastic properties for 0°, 45° and 90° samples.

Test Methods	Tested at 15 °C; 5% Compressive Strain			Tested at Af + 10 °C; 7% Compressive Strain		
Samples	0°	45°	90°	0°	45°	90°
Recoverable strain (%)	1.8	3.2	2.2	5.42	3.06	5.83
Irrecoverable strain (%)	3.2	1.8	2.8	1.58	3.94	1.17
Recovery ratio (%)	36	64	44	77.4	43.7	83.3

**Table 4 micromachines-14-01711-t004:** Comparison of the optimal superelastic properties (tested at Af + 10 °C) in various LPBFed NiTi.

Reference (Tested at Af + 10 °C)	Recoverable Strain (%)	Recovery Ratio (%)
[39]	5.62	98
[40]	5.5	94.8
[41]	5.32	93.2
This work	5.83	83.3

## Data Availability

The data presented in this study are available on request from the corresponding author. The data are not publicly available due to privacy.

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
