# Peer review of "Effects of Building Directions on Microstructure, Impurity Elements and Mechanical Properties of NiTi Alloys Fabricated by Laser Powder Bed Fusion"

_micromachines, 2023, doi:10.3390/mi14091711_

Round 1

Reviewer 1 Report

In this paper, the relationship between building direction and material property including macro and micro was studied. Various research contents were explored, such as microstructure, impurity elements, etc. The paper has a certain amount of work, but there are still the following problems.

1. In the abstract, the motivation behind this paper is that other research on NiTi alloys is not comprehensive enough, rather than that this research direction has significant engineering or scientific value.

2. The abstract lacks logical relationships and only puts the experimental results together. The conclusion you mention “which may affect the transformation temperature by changing Ni/Ti ratio” seems uncertain.

3. In the second paragraph on page four, you didn’t explain where the elements H and S in specimens come from.

4. On page seven, the conclusion of the last paragraph, just indicates the consistency of experiment results between EBSD and TEM methods. And the last paragraph on page eight also has the same problem. Lack of real conclusions or discussions about your results.

5. In the conclusion, you should give some advice about how to choose the appropriate building direction based on the different requirements for the engineering application value of your paper.

6. In Figure 11, I don’t know whether the stress and strain are engineering parameters or true parameters, you didn’t make that clear.

Some minor concerns:

1. In your abstract, “Af” is the first abbreviation you mentioned, so you must explain the meaning of it.

2. Table 1 is not needed because all of the print parameters are the same.

3. On page 7, “In terms of the martensite phases, more martensite phases were observed in 0° sample and 90° sample than 45° sample.”, this sentence is a little confusing, please check the full text for syntax problems.

4. Reference [17,18] should be in front of “.” on page 4.

5. All of the figures in your paper should be clearer instead of using screenshots, especially Figure 2, figure 7(b).

6. Figure 9 and Table 2 express the same information.

Overall, the paper shows promise, but there are areas that could be improved to enhance the clarity and coherence of the writing.

Firstly, there are several instances where the choice of vocabulary could be refined. In some cases, the use of more precise and appropriate terms would better convey the intended meaning. Additionally, the overall flow of the sentences could be improved by using transition words and phrases to connect ideas more effectively.

Secondly, attention should be paid to sentence structure and grammatical accuracy. There are instances where subject-verb agreement is not consistent, and some sentences are overly complex and difficult to follow. Simplifying and rephrasing these sentences would greatly improve readability.

Furthermore, I noticed several punctuation errors throughout the manuscript. Careful proofreading is needed to correct these mistakes, as they can affect the overall comprehension of the text.

Lastly, it is important to ensure that the paper adheres to the appropriate academic writing style and conventions. Consistency in formatting, citation, and referencing is crucial for maintaining scholarly integrity.

In conclusion, I recommend that the author further improve the English language quality and grammar of the manuscript. By addressing the issues mentioned above, the paper will become more polished and accessible to readers.

Author Response

Our response is linked to the attachment.

Reviewer 2 Report

See attached.

See attached.

Author Response

Our response is linked to the attachment

Reviewer 3 Report

The manuscript entitled “micromachines-2511232” dealing with AM has been reviewed. The paper has been nicely written but needs significant improvement. Please follow my comments.

1.     Use laser-based powder bed fusion instead of SLM. Follow ASTM 52900. This should be done in section 2.1.

2.     How did you select your process parameters in Table 1. Please add a sentence about this.

3.     Figure 1 needs scale bar for the printed components.  

4.     What is the main issue that will be solved by this investigation? Please clarify it in the text.

5.     Please add a brief statement on your methodology.

6.     Please proofread the paper.

7.     AM has many usages in different industries. To highlight your work, add a short note in the introduction by using the following papers and mention the privilege of lasers in manufacturing. “Benchmark models for conduction and keyhole modes in laser-based powder bed fusion of Inconel 718”. “Comparative study on the properties of 17-4 PH stainless steel parts made by metal fused filament fabrication process and atomic diffusion additive”.

Needs some improvement. 

Author Response

Our response is shown in the attachment.

Round 2

Reviewer 2 Report

You misunderstood the following comment, and you should revise the item of "vertical direction" in the abstract as "vertical direction (VD)".

4. The shortage items of vertical direction, VD, should be given in the abstract, instead of in the sentence of “As shown in Figure 4(a), the inverse pole figure (IPF) of the 0° sample along the RD-TD direction shows a clear solidification texture of <100>∥TD along the vertical direction (VD).”

Author Response

Thanks, I've made the changes in the abstract and the text involved and marked it red.